# Unlocking 2D Promptable Foundation Models for 3D Vessel Segmentation by Automatic Prompt Generation

**Ziyu Zhang**[*1]   iD                     ZHANGZIYU@NJU.EDU.CN
[1] *Nanjing University*

**Yi Yu**[*2]   iD                          YI.YU@OSUMC.EDU
**Yuan Xue**[2]   iD                      YUAN.XUE@OSUMC.EDU
[2] *The Ohio State University*

**Editors:** Accepted for publication at MIDL 2026

## Abstract

3D vessel segmentation is a core task in medical image analysis, playing a crucial role in disease diagnosis and surgical planning. While fully supervised 3D segmentation methods rely on costly high-quality annotations, promptable models (e.g., ScribblePrompt) provide a promising alternative with their zero-shot generalization capability for efficient 3D segmentation. Nevertheless, when directly applied to 3D tasks, these 2D methods require slice-wise prompts, disregarding the continuity of 3D structures and leading to low efficiency. To address this issue, we propose an innovative method based on automatic prompt generation, which integrates with pre-trained 2D interactive models to achieve efficient 3D vessel segmentation. By leveraging spatial continuity and contextual information, our method automatically generates prompts across the entire 3D volume from a single user-provided prompt. Experiments conducted on public and in-house vessel datasets demonstrate the effectiveness of the proposed method, showing that it achieves segmentation accuracy comparable to or better than state-of-the-art models, while significantly reducing the interaction cost.

**Keywords:** Vessel Segmentation, Foundation Models, Automatic Prompting, Topology Preservation.

## 1. Introduction

Cardiovascular diseases remain the leading cause of death worldwide. According to data from the World Health Organization, approximately 17.9 million people die from cardiovascular diseases each year, accounting for 32% of global deaths (Chen et al., 2020). Accurate vascular morphometry analysis is crucial for early diagnosis, risk assessment, and treatment planning for cardiovascular diseases(Sweeney et al., 2024; Zeng et al., 2024).

Vascular segmentation, as a fundamental technology in medical image analysis, aims to accurately extract vascular structure from medical images, including vessel trajectory, morphology, diameter, and other clinical biomarkers(Liu et al., 2022; Qi et al., 2023). These details are vital for diagnosing vascular conditions such as stenosis or aneurysms and are valuable for surgical planning and intervention guidance(Fu et al., 2023; Yao et al., 2023). However, significant challenges persist due to factors such as the intricate complexity of vascular structures, inherent imaging quality limitations, and various pathological alterations that complicate automated analysis (Acebes et al., 2024; Shi et al., 2024).

---

[*] Contributed equally

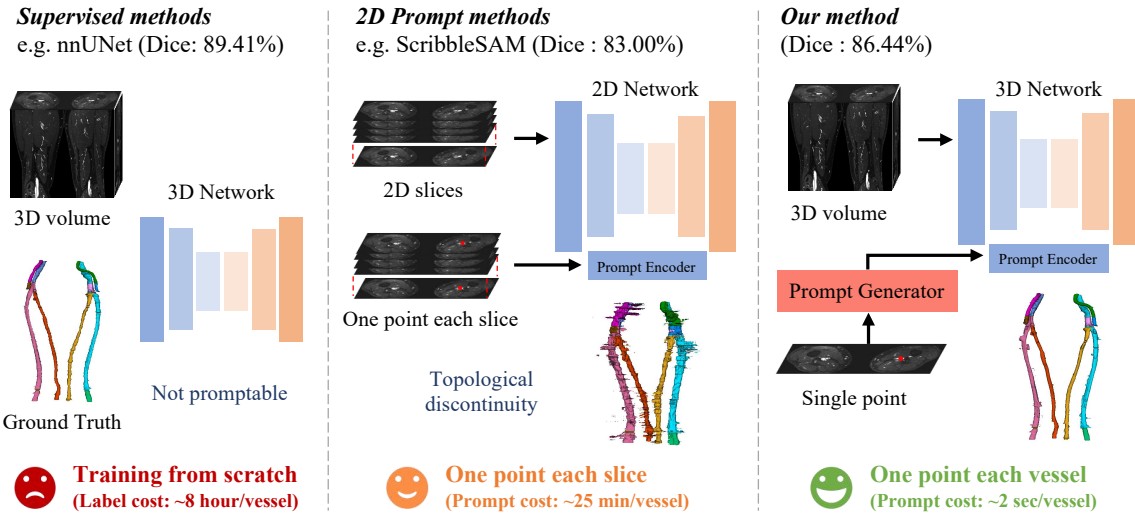

Figure 1: Motivation and Performance Overview. Comparison of three paradigms for 3D vessel segmentation. (Left) Fully Supervised Methods (e.g., nnU-Net): Achieve high accuracy but require labor-intensive voxel-wise annotations (∼8 hours/vessel). (Middle) Naive 2D Prompt Methods (e.g., ScribblePrompt): Apply slice-by-slice prompting, which is still time-consuming (∼25 min/vessel) and often leads to topological discontinuities (broken vessels) due to lack of 3D context. (Right) Our Method: By utilizing a single initialization point, our approach automatically generates prompts via topological continuity learning. We achieve competitive performance comparable to supervised methods but with orders of magnitude faster interaction (∼2 sec/vessel), ensuring coherent 3D structures.

Traditional manual segmentation is labor-intensive and prone to inter-observer variability. While deep learning-based methods have automated segmentation in mainstream modalities like coronary CTA (Dai et al., 2024; Geng et al., 2024; Qiu et al., 2023; Xie et al., 2024), their success heavily relies on large-scale, high-quality annotated datasets (Guo et al., 2024). This data dependency becomes a bottleneck for emerging imaging techniques such as Fe-MRA or underrepresented peripheral vessels (Ghodrati et al., 2022). Recently, promptable foundation models, such as the Segment Anything Model (SAM), have emerged as a promising solution to annotation scarcity due to their impressive zero-shot generalization capabilities (Zhang et al., 2025a,b). However, a critical limitation hinders their application in 3D medical imaging: standard foundation models are inherently 2D. Applying them to 3D volumes typically requires slice-by-slice prompting, which disrupts the 3D spatial continuity of vascular structures and results in prohibitive interaction costs (Magg et al., 2025). This issue is particularly pronounced in complex scenarios like Fe-MRA, where sparse acquisition protocols lead to topological discontinuities (Si et al., 2025), making simple slice-wise propagation unreliable (Si et al., 2024).

To address the dimensionality gap, recent efforts have sought to adapt foundation models for 3D medical tasks, generally falling into training-free or trainable categories (Zhang

et al., 2025b). Training-free methods, such as $\mu$SAM (Archit et al., 2025a) and MedicoSAM (Archit et al., 2025b), typically employ projection-based strategies where masks are projected to adjacent slices to derive prompts. Some approaches also attempt to leverage image registration to align adjacent slices for label propagation. However, registration-based methods are often computationally expensive and prone to failure when dealing with non-rigid deformations or abrupt topological changes common in vascular structures. These approaches often lack inherent 3D context and rely heavily on the assumption of minimal inter-slice variation, leading to error accumulation in datasets with anisotropic resolutions or large spacing. Conversely, trainable adaptations like MedSAM2 (Ma et al., 2025) and PAM (Chen et al., 2025) introduce memory banks or cross-attention mechanisms to capture volumetric context. While effective, MedSAM2 treats static 3D volumes as temporal video sequences, incurring significant computational overhead due to heavy memory modules. Similarly, PAM's slice-to-slice attention mechanism can struggle with long-range dependencies or abrupt topological changes that are not captured by immediate neighbors. Consequently, there remains a need for an efficient, topology-aware strategy that explicitly models anatomical spatial consistency without the heavy computational burden of video-based architectures.

To bridge the gap between 2D foundation models and 3D vascular segmentation, we propose an innovative framework that unlocks the potential of SAM through *automatic prompt generation*. As illustrated in Fig. 1, our approach addresses the inefficiency of slice-wise interaction by leveraging the inherent geometric continuity of vessels. Instead of requiring dense prompts for every slice, our method necessitates only a single initial point. It then employs a topology-aware strategy to automatically propagate prompts across the entire 3D volume, effectively stitching 2D predictions into a coherent 3D structure. This strategy integrates two key innovations: (1) an end-to-end segmentation framework with global and local continuity constraints to overcome spatial inconsistencies; and (2) a confidence-aware prompt generation mechanism that exploits vascular topology to refine and extend segmentation cues iteratively. By combining the generalization power of SAM with explicit vascular priors, we reduce the annotation time from hours to seconds while maintaining high topological consistency.

The main contributions are summarized as follows:

- We introduce a novel topology-constrained framework that extends 2D promptable models to 3D vascular segmentation, requiring only a single initialization point to achieve comprehensive volumetric segmentation.

- We develop a topology-driven automatic prompt generation strategy that leverages vascular structural continuity with iterative confidence-aware refinement, significantly reducing interaction costs compared to slice-wise prompting.

- Extensive experiments demonstrate the effectiveness of our method, achieving state-of-the-art performance with 86.44% Dice on a public CTA dataset and 80.20% Dice on an in-house Fe-MRA dataset, showcasing superior generalization across different modalities and vascular complexities.

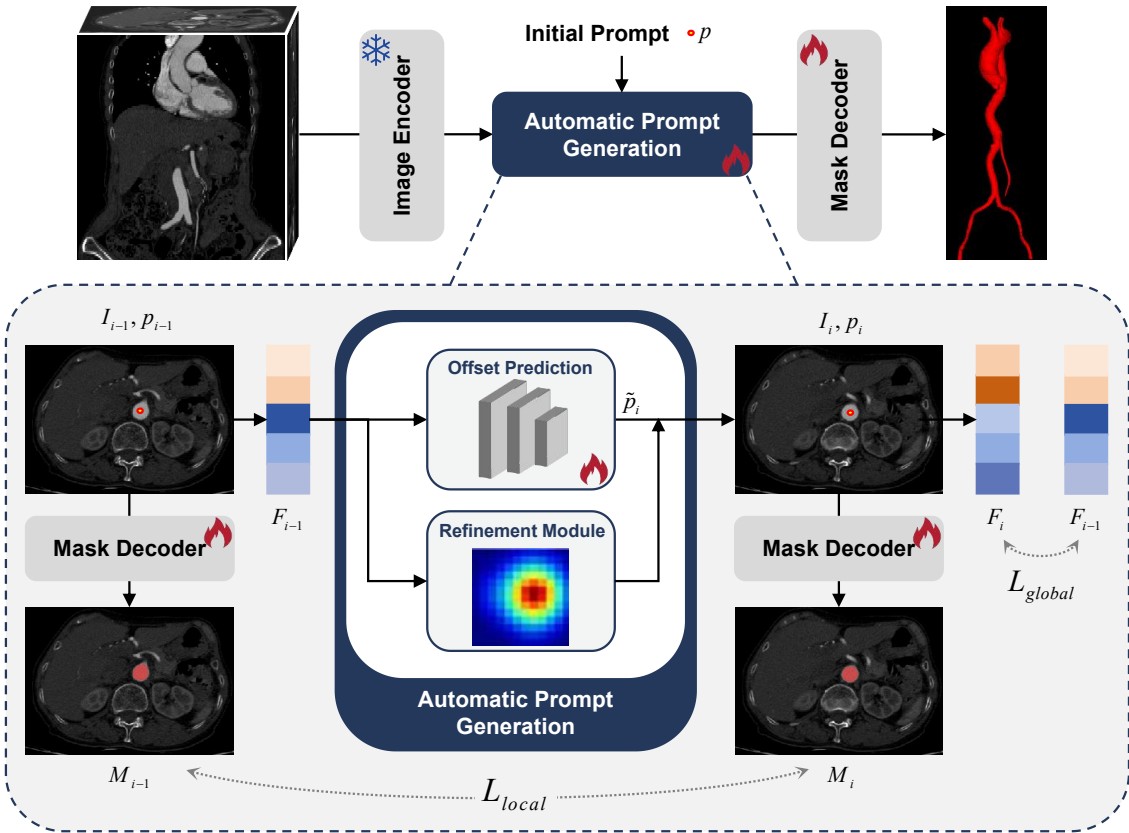

Figure 2: Overview of the proposed framework. (Top) The slice-wise framework with 3D consistency processes the volume slice-by-slice, enforcing volumetric consistency via Local Continuity Loss ($L_{local}$) and Global Consistency Loss ($L_{global}$). (Bottom) The Automatic Prompt Generation module propagates the prompt from slice $i-1$ to $i$. It utilizes an Offset Prediction network and a Refinement Module based on previous features $F_{i-1}$ to accurately locate the vessel center $p_i$.

## 2. Method

Upon the success of prompt-based learning in medical imaging (Ma et al., 2024), we propose a dimension-hybrid framework that unlocks the potential of 2D foundation models for 3D vascular segmentation. As illustrated in Fig. 2, our method requires only a *single* user-provided point on the first slice to segment the entire 3D volume. The framework consists of two synergistic components: (1) a **Slice-wise Framework with 3D consistency** that extends a 2D SAM-based backbone with local and global geometric constraints to ensure volumetric consistency, and (2) a **Topology-Aware Prompt Generator** that automatically propagates and refines point prompts across slices, eliminating the need for slice-by-slice prompting.

### 2.1. Weakly Supervised Slice-wise Framework with 3D Geometric Constraints

Conventional volumetric methods (Isensee et al., 2021) often suffer from high computational costs, while naive 2D slice-based approaches neglect the inherent 3D continuity of vascular structures. Our approach bridges this gap by employing a pre-trained 2D SAM encoder to extract robust features from individual slices, while enforcing 3D consistency through a dual-level regularization framework.

Given a 3D volume, we process it as a sequence of 2D slices. For each slice $I_i$, the network takes the image and an automatically generated prompt $p_i$ (detailed in Sec. 2.2) to predict a segmentation probability map $P_i$. To address the topological fragmentation common in 2D-based predictions, we introduce two loss functions:

**Local Continuity Loss ($L_{local}$).** Vascular structures exhibit gradual morphological transitions across adjacent slices. To enforce this smoothness, we impose a local continuity constraint that penalizes abrupt changes in the segmentation shape and position. We formulate the inter-slice gradient consistency as:

$$L_{grad} = \sum_i \|\nabla_z P_i - \nabla_z P_{i+1}\|_1 \tag{1}$$

where $\nabla_z P_i = P_i - P_{i-1}$ represents the discrete difference of probability maps along the $z$-axis (slice dimension). Minimizing the difference between consecutive gradients ($\nabla_z P_i$ and $\nabla_z P_{i+1}$) effectively encourages a constant velocity in shape evolution, resulting in smooth vascular boundaries.

Additionally, to ensure intra-slice spatial smoothness and reduce noise, we incorporate a Total Variation (TV) term. The aggregate local loss is defined as:

$$L_{local} = L_{grad} + \lambda_{\text{TV}} \sum_i \|\nabla_{xy} P_i\|_1 \tag{2}$$

where $\nabla_{xy}$ denotes the spatial gradient within the 2D slice, and $\lambda_{\text{TV}}$ balances the regularization strength (empirically set to 0.1).

**Global Consistency Loss ($L_{global}$).** While local constraints handle immediate transitions, they may fail to prevent semantic drift over long distances. We therefore introduce a global consistency loss operating in the latent feature space. We assume that the deep feature representations of the same vessel should remain semantically consistent across the volume. We maximize the cosine similarity between the feature embeddings $F$ of adjacent slices:

$$L_{global} = \sum_i \left(1 - \frac{F_i \cdot F_{i+1}}{\|F_i\|_2 \|F_{i+1}\|_2}\right) \tag{3}$$

where $F_i$ is the bottleneck feature map extracted by the Prompt Encoder for slice $i$. This constraint ensures that the network maintains a stable internal representation of the vascular target throughout the 3D volume.

### 2.2. Topology-Aware Automatic Prompt Generation

Standard prompt-based methods require manual interaction for each slice, which is prohibitive for 3D volumes with hundreds of slices. To automate this, we design a **Point**

**Prompt Generator** (Fig. 2, bottom) that predicts the optimal prompt $p_i$ for the current slice based on the segmentation result of the previous slice. It operates in two steps:

***Step 1:*** **Feature-Driven Offset Prediction.** Vessels are continuous tubular structures. The center of a vessel in slice $i$ can be inferred from its position in slice $i - 1$ plus a displacement vector. We employ a lightweight Offset Prediction Module that utilizes the deep features $F$ (from the Prompt Encoder) to estimate this transition. The tentative prompt $\tilde{p}_i$ is calculated as:

$$\tilde{p}_i = p_{i-1} + \text{Net}_{\text{offset}}(F_{i-1}) \tag{4}$$

where $p_{i-1}$ is the prompt used in the previous slice. By conditioning the offset on the rich semantic features $F_{i-1}$, the network learns to anticipate the vessel's trajectory (e.g., curvature and branching).

***Step 2:*** **Confidence-Guided Refinement.** Relying solely on offset prediction may lead to accumulated errors over time. To correct this, we refine the prompt using the network's own confidence map. After obtaining the initial probability map $P_i$ using $\tilde{p}_i$, we identify the high-confidence region $\arg\max(P_i)$. The final refined prompt $p_i$ is computed as a weighted fusion:

$$p_i = (1 - \lambda) \cdot \tilde{p}_i + \lambda \cdot \arg\max(P_i) \tag{5}$$

where $\lambda \in [0.1, 0.5]$ is an adaptive scalar derived from the peak confidence value of $P_i$. This refinement step acts as a self-correction mechanism: if the network is confident in its segmentation (high $\lambda$), the prompt is pulled towards the actual vessel center; if uncertainty is high, the system relies more on the topological prior $\tilde{p}_i$. This ensures robust tracking even in complex vascular bifurcations.

## 3. Experiments

### 3.1. Dataset

We evaluate the efficacy and robustness of our proposed method on two distinct vascular datasets: an in-house Fe-MRA dataset and the publicly available SEG.A. 2023 challenge dataset (Radl et al., 2022). The Fe-MRA dataset serves to assess performance under weakly supervised conditions for complex peripheral vessels, while the public dataset is employed to rigorously validate cross-modality generalization and segmentation accuracy on the Aortic Vessel Tree in CTA scans.

**Fe-MRA Dataset.** The in-house Fe-MRA dataset was retrospectively collected from Nanjing Drum Tower Hospital, The Affiliated Hospital of Nanjing University Medical School, between 2023 and 2025, under an IRB-approved protocol. All patient data were anonymized prior to analysis. It consists of 50 cases exhibiting a wide range of vascular morphologies, including normal vessels, lower extremity varicose veins, and arterial thrombosis. This diversity introduces significant challenges for segmentation, such as severe tortuosity and vessel occlusion. The images are whole-body MRA scans acquired 30 minutes post-injection of Ferumoxytol. To ensure data diversity, scans were performed using two different MRI systems: a 1.5T Philips scanner and a 3.0T GE scanner. The dataset features heterogeneous spatial resolutions, with voxel spacing ranging from $0.3 \times 0.3 \times 1.0$ mm to $0.8 \times 0.8 \times 1.0$ mm, capturing key lower extremity vessels such as the femoral artery/vein and great saphenous vein. The ground truth was established through a rigorous two-stage process: initial dense

annotation by two junior radiologists, followed by a final review and revision by a senior radiologist to ensure clinical accuracy.

**SEG.A. Dataset.** To rigorously evaluate generalization, we further validate our method on the SEG.A. benchmark from the MICCAI 2023 Challenge. This dataset comprises 56 CTA scans of the Aortic Vessel Tree aggregated from three distinct centers, introducing significant anatomical and scanner variability. The data exhibits high heterogeneity in resolution, with voxel spacing ranging from $0.44 \times 0.44 \times 0.50$ mm to $1.37 \times 1.37 \times 5.00$ mm. Notably, the dataset includes large-scale volumes exceeding 1,000 slices along the Z-axis, providing a challenging testbed for processing extensive 3D vascular structures.

**Data Splitting.** To ensure a rigorous evaluation and prevent data leakage, we strictly perform data splitting at the patient (volume) level, rather than the slice level. For the Fe-MRA dataset, the 50 patients are divided into Training (30), Validation (5), and Testing (15). For the SEG.A. dataset, we adhere to the official challenge split guidelines.

### 3.2. Implementation Details and Evaluation Metrics

**Architecture and Initialization.** Our framework is built upon the architecture of ScribblePrompt-SAM (Wong et al., 2024). Specifically, we utilize the ViT-b backbone as the Image Encoder. Given that ScribblePrompt-SAM has been extensively pre-trained on a large-scale collection of medical images, we keep the Image Encoder completely frozen during training to preserve its robust feature extraction capabilities and maintain computational efficiency. Similarly, the standard Prompt Encoder remains frozen as its positional encodings for point prompts do not require domain-specific adaptation. Conversely, the Mask Decoder is initialized from ScribblePrompt-SAM but is set to be fully trainable. This allows the model to adapt to the specific topology of thin, tubular vascular structures, which may differ from the general targets seen during pre-training. The mask decoder is optimized with $L_{local}$ which enforces continuous vessel boundaries across neighboring slices and penalizes abrupt shape changes. Finally, our core contribution, the automatic prompt generator, is a lightweight MLP that is fully trainable and initialized randomly. The automatic prompt generator is optimized by $L_{global}$ to learn semantic consistency of vessel-center representations across slices, guiding it to track the same anatomical structure. All models were implemented in PyTorch and trained on a single NVIDIA RTX 4090 GPU (24GB). Training spanned 200 epochs using the AdamW optimizer (learning rate: $1 \times 10^{-4}$, weight decay: $1 \times 10^{-5}$) and a cosine annealing scheduler with a 10-epoch warm-up.

**Training Protocol.** We randomly sample pairs of adjacent slices $(I_{i-1}, I_i)$ from the training volumes in each iteration. The model utilizes the features and predictions from the previous slice $I_{i-1}$ to predict the offset and generate prompts for the current slice $I_i$. This strategy forces the automatic prompt generator to learn robust slice-to-slice transition logic rather than memorizing specific volumetric positions. During the backward pass, gradients derived from the segmentation loss and our proposed 3D consistency losses are back-propagated exclusively to update the automatic prompt generator and the Mask Decoder.

**Evaluation Metrics.** We employ a comprehensive set of metrics to assess both geometric accuracy and topological fidelity. Standard segmentation performance is measured using the Dice Similarity Coefficient (DSC) and the 95% Hausdorff Distance (HD95). Given the tubular nature of vascular structures, we specifically include clDice (Shit et al., 2021) to

Table 1: Quantitative comparison on the public SEG.A. segmentation task. Supervised methods serve as the upper bound. Best results among point-prompted methods are highlighted in **bold**.

| Type | Method | Dice ($\% \uparrow$) | clDice ($\% \uparrow$) | HD95 ($mm \downarrow$) | $\beta_0$ error ($\downarrow$) |
|------|--------|------|--------|------|---------|
| Supervised | UNet | 82.08 | 92.56 | 18.26 | 0.14 |
| | nnU-Net | 89.41 | 96.29 | 9.37 | 0.09 |
| Point Prompt | SAMed-2D | 81.21 | 79.23 | 49.41 | 155.33 |
| | MIDeepSeg | 80.91 | 71.56 | 38.08 | 116.28 |
| | ScribblePrompt-UNet | 81.55 | - | 42.39 | - |
| | **Ours(ScribblePrompt-UNet)** | 82.72 | - | 35.06 | - |
| | ScribblePrompt-SAM | 83.00 | 85.65 | 26.42 | 3.26 |
| | **Ours(ScribblePrompt-SAM)** | **86.44** | **89.83** | **19.46** | **1.88** |

evaluate centerline connectivity and Betti Error to quantify topological consistency (e.g., broken vessels or false loops), ensuring clinical reliability.

### 3.3. Experimental Results

To comprehensively evaluate the clinical viability and technical superiority of our point-prompted vascular segmentation framework, we conducted a multi-dimensional comparison against two distinct methodological paradigms. The first category represents the **fully supervised baselines**, including the classic UNet (Ronneberger et al., 2015) and the self-configuring nnU-Net (Isensee et al., 2021), which require labor-intensive pixel-level annotations. The second category comprises **state-of-the-art weakly supervised and interactive methods**, specifically SAMed-2D (Cheng et al., 2023), MIDeepSeg (Luo et al., 2021), and ScribblePrompt (Wong et al., 2024). Our analysis focuses not only on volumetric overlap (Dice) but places particular emphasis on topological fidelity (clDice, $\beta_0$ error), which is critical for downstream vascular analysis such as centerline extraction and hemodynamic simulation.

**Performance on Public SEG.A. Dataset.** Table 1 details the quantitative performance on the SEG.A. dataset. Our method demonstrates a commanding lead among weakly supervised approaches, effectively bridging the gap towards fully supervised baselines. We achieve a Dice score of **86.44%**, surpassing the closest competitor, ScribblePrompt, by **3.44%**. Notably, this performance is highly competitive with the fully supervised nnU-Net (89.41%), suggesting that our point-prompted strategy can recover the majority of vascular structures with significantly reduced annotation costs. The HD95 metric, which is sensitive to outliers, is reduced to **19.46** $mm$ in our method. This indicates that our approach effectively suppresses false positives in the background, a common issue in SAM-based adaptations where the model struggles with low-contrast medical boundaries. Most critically, our method excels in preserving vascular connectivity. We outperform ScribblePrompt by **4.18%** in clDice and reduce the Betti number error ($\beta_0$) from 3.26 to **1.88**. This low topological error rate implies that our segmentation masks contain significantly fewer fracture points, ensuring a continuous vascular tree structure that is essential for clinical diagnosis.

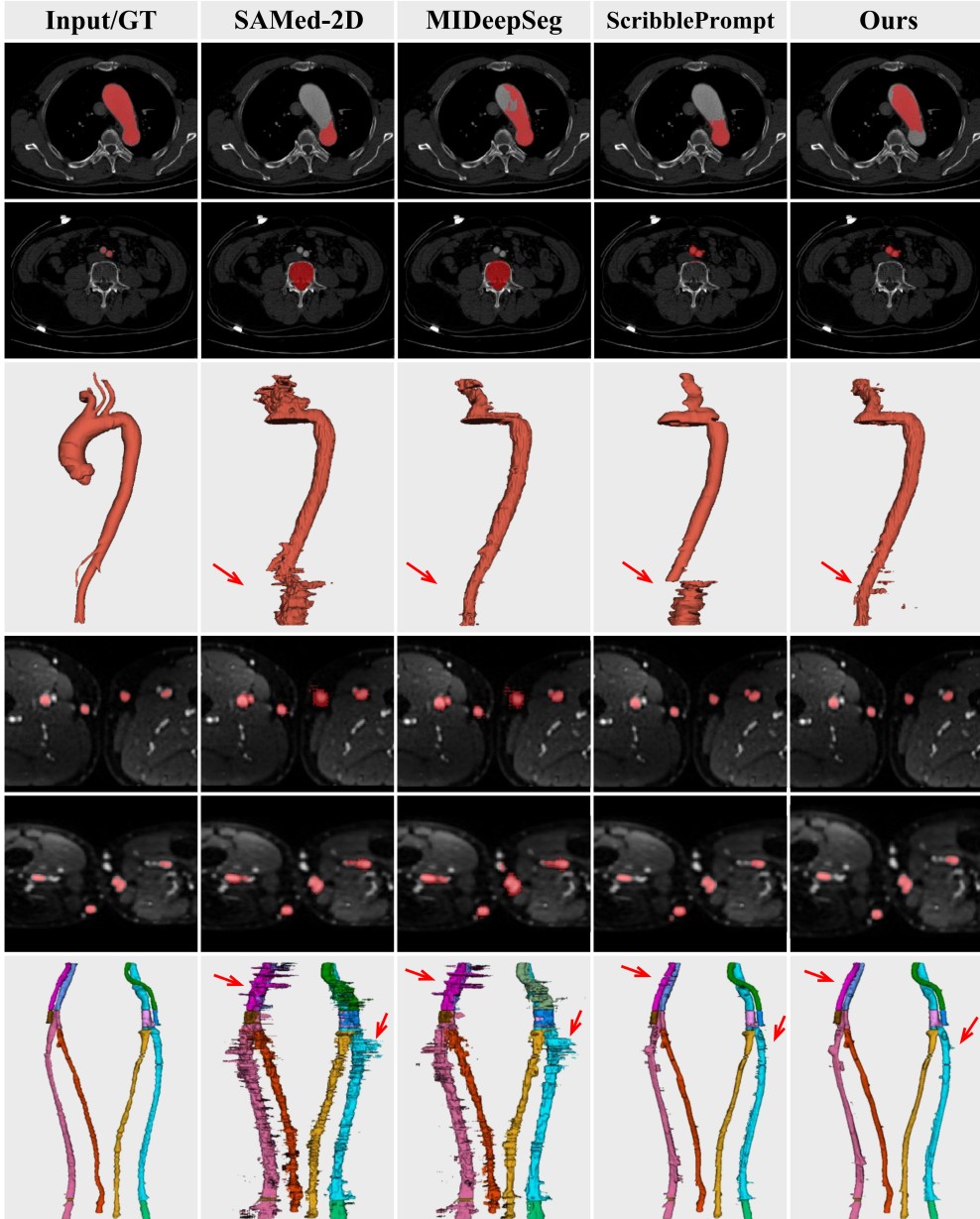

Figure 3: Qualitative comparison of vascular segmentation on SEG.A. (top 3 rows) and Fe-MRA (bottom 3 rows). Topological breaks or false positives are observed in baseline methods, whereas our method effectively corrects them.

**Robustness on In-house Fe-MRA Dataset.** The Fe-MRA dataset represents a significantly more challenging scenario characterized by intricate peripheral vessels and variable contrast-to-noise ratios. As shown in Table 2, this domain shift exposes the fragility of existing methods. Methods like SAMed-2D and MIDeepSeg exhibit a drastic performance

Table 2: Quantitative comparison on the challenging in-house Fe-MRA dataset. Note the significant improvement in topological metrics (clDice and $\beta_0$ error) achieved by our method.

| Method | Dice (% ↑) | clDice (% ↑) | HD95 ($mm$ ↓) | $\beta_0$ error (↓) |
|---|---|---|---|---|
| SAMed-2D | 69.57 | 61.65 | 38.69 | 207.6 |
| MIDeepSeg | 72.99 | 67.17 | 38.71 | 172.3 |
| ScribblePrompt | 77.25 | 86.21 | 19.92 | 67.46 |
| **Ours** | **80.20** | **88.13** | **11.18** | **40.25** |

drop (Dice $< 72\%$). These models, primarily trained on natural images or standard medical datasets, lack the specific inductive bias required to track thin, branching vessels, leading to fragmented outputs (high $\beta_0$ error $> 172$). In contrast, our framework maintains high robustness with a Dice score of **80.20%**. More importantly, we achieve a clDice of **88.13%**, which is comparable to the performance on the simpler SEG.A. dataset. This demonstrates that our slice-to-slice propagation mechanism effectively utilizes the 3D coherence of blood vessels, allowing the model to trace vessels even when the local contrast is weak. The $\beta_0$ error of our method (**40.25**) is substantially lower than ScribblePrompt (67.46). This reduction confirms that our approach is less prone to "breaking" thin vessel segments, a frequent artifact in slice-wise 2D segmentation methods that ignore inter-slice consistency. **Qualitative Analysis.** Fig. 3 provides a visual comparison that corroborates our quantitative findings. In the SEG.A. dataset (top 3 rows), baseline methods often struggle with boundary definition. For instance, SAMed-2D tends to under-segment the vessel walls, while ScribblePrompt occasionally leaks into adjacent tissues. The disparity is even more pronounced in the Fe-MRA dataset (bottom 3 rows). As indicated by the red arrows, competing methods produce discontinuous dotted patterns for distal vessels, severing the vascular tree. Our method, leveraging the propagated point prompts, successfully reconstructs the complete vascular geometry. The 3D renderings clearly show that our result is the only one that maintains the structural integrity of the entire vascular network without significant topological breaks.

### 3.4. Ablation Study

To investigate the individual contributions of the proposed components to the overall segmentation performance and topological consistency, we conducted a comprehensive ablation study on the SEG.A. dataset. We established a baseline model using the pre-trained SAM-based 2D network with slice-by-slice manual prompting (simulated by using the ground truth center of the previous slice without any learnable offset or refinement). As shown in Table 3, we progressively incorporated the Local Continuity Loss ($\mathcal{L}_{local}$), Global Consistency Loss ($\mathcal{L}_{global}$), Offset Prediction, and Confidence-Guided Refinement.

**Effectiveness of Geometric Constraints.** The baseline model, which treats 3D segmentation as independent 2D tasks, achieved a Dice score of $81.21\%$. Introducing the Local Continuity Loss ($\mathcal{L}_{local}$) significantly improved the boundary smoothness between slices, yielding a $1.84\%$ increase in Dice and reducing the HD95 by over 10mm. This confirms that

Table 3: Ablation study on the SEG.A. dataset. We progressively add components to the Baseline (Naive Slice-wise SAM). $\mathcal{L}_{local}$: Local Continuity Loss; $\mathcal{L}_{global}$: Global Consistency Loss; *Offset*: Feature-Driven Offset Prediction; *Refine*: Confidence-Guided Refinement.

| Baseline | $\mathcal{L}_{local}$ | $\mathcal{L}_{global}$ | Offset | Refine | Dice (% ↑) | clDice (% ↑) | HD95 (mm ↓) | $\beta_0$ error (↓) |
|:---:|:---:|:---:|:---:|:---:|:---:|:---:|:---:|:---:|
| ✓ | | | | | 81.21 | 79.23 | 49.41 | 155.33 |
| ✓ | ✓ | | | | 83.05 | 82.05 | 38.12 | 89.45 |
| ✓ | ✓ | ✓ | | | 84.19 | 85.12 | 25.60 | 24.18 |
| ✓ | ✓ | ✓ | ✓ | | 85.33 | 88.51 | 21.05 | 8.62 |
| ✓ | ✓ | ✓ | ✓ | ✓ | **86.44** | **89.83** | **19.46** | **1.88** |

penalizing inter-slice gradient inconsistencies effectively mitigates the "stacking artifacts" common in 2D-to-3D adaptation. Furthermore, the addition of the Global Consistency Loss ($\mathcal{L}_{global}$) provided a substantial boost in topological fidelity, increasing clDice from 82.05% to 85.12%. By enforcing feature-level similarity across the volume, $\mathcal{L}_{global}$ prevents semantic drift in slices with low contrast or noise, ensuring the model maintains a stable representation of the vessel throughout the scan.

**Impact of Automatic Prompt Generation Strategies.** A critical innovation of our framework is the replacement of manual prompts with an automated mechanism. We first evaluated the Offset Prediction module (Eq. 4), which predicts the vessel center displacement based on previous features. This mechanism alone achieved a Dice of 85.33%, demonstrating that the network can effectively learn the trajectory of vascular structures. However, relying solely on offset prediction led to accumulated errors in tortuous vessel segments, as indicated by a suboptimal $\beta_0$ error. Finally, incorporating the Confidence-Guided Refinement (Eq. 5) resulted in the best performance across all metrics. This module acts as a self-correction mechanism; by pulling the prompt towards high-confidence regions, it recovered 1.32% in clDice and reduced the topological error ($\beta_0$) to 1.88. This result highlights the necessity of combining historical trajectory priors (offset) with current observational evidence (confidence map) for robust 3D tracking.

**Impact of Propagation Strategies.** To investigate the optimal strategy for propagating vessel information across slices, we conducted an ablation study comparing our method with registration baselines and the prompt-based $\mu$SAM model (Archit et al., 2025a). The results are presented in Table 4. For the registration baselines, we applied Rigid and Affine transformations to propagate the mask from the previous slice to the current slice. We evaluated two point anchors: the geometric center and centroid of the vessel mask. As observed, centroid-based anchoring is more effective for affine registration, although it does not improve the rigid baseline. This confirms that tracking the anatomical trajectory of the vessel is crucial. However, even the best-performing baseline (Affine-Centroid) yields a lower Dice score compared to our method. This limitation stems from the fact that vessels undergo complex, non-linear morphological deformations across slices that simple rigid or affine transformations cannot model. While $\mu$SAM offers a simple, training-free solution by directly projecting masks from adjacent slices, its accuracy is limited (83.33%) due to the

Table 4: Ablation study on comparison of different slice-to-slice propagation strategies. We compare our proposed method against classical registration method and the recent training-free model $\mu$SAM. Rigid and Affine denote using the center or centroid point from the previous slice transformed by rigid or affine registration as the prompt for the current slice. Our method achieves the highest segmentation accuracy with significantly lower inference latency.

| Method | Dice (% ↑) | Inference Time (s/slice) |
|---|---|---|
| Rigid (center) | 59.29 | 0.55 |
| Rigid (centroid) | 54.21 | 0.55 |
| Affine (center) | 82.57 | 0.56 |
| Affine (centroid) | 84.42 | 0.56 |
| $\mu$SAM | 83.33 | - |
| Ours | 86.44 | 0.10 |

lack of deformation modeling. Our method, which explicitly learns inter-slice relationships, achieves a superior Dice score of 86.44%.

## 4. Conclusion

In this paper, we presented a novel framework that adapts the Segment Anything Model for 3D volumetric vessel segmentation. By integrating a geometric continuity constraint and a global consistency mechanism, we effectively resolved the slice-to-slice inconsistency problem inherent in 2D-to-3D adaptation. Moreover, our proposed learnable offset prediction and confidence-guided refinement modules eliminate the need for manual interaction, enabling fully automated and robust vessel tracking. Extensive experiments demonstrate that our method outperforms existing state-of-the-art approaches in both segmentation accuracy and topological fidelity. We believe this work provides a scalable and effective solution for leveraging 2D foundation models in 3D medical image analysis.

**Limitations and Future Work.** Although the refinement module reduces error propagation, the slice-by-slice processing within our model remains inherently susceptible to it. Meanwhile, the slice-by-slice processing also prevents the method from recovering vessel segmentation that turns back into earlier slices. Additionally, handling complex bifurcations where a single vessel splits into multiple branches remains a challenge for the current single-point prompting mechanism. Future work will focus on integrating a multi-hypothesis tracking approach to handle bifurcations more effectively and exploring bi-directional propagation to further reduce error accumulation.

## Acknowledgments

Ziyu Zhang was supported in part by the Natural Science Foundation of Jiangsu Province of China (BK20241202), the China Postdoctoral Science Foundation (2024M751394, GZC2024 0691), and the Jiangsu Funding Program for Excellent Postdoctoral Talent (2024ZB433).

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
