# OpenReview forum: "Unlocking 2D Promptable Foundation Models for 3D Vessel Segmentation by Automatic Prompt Generation"
_MIDL.io/2026/Conference — MIDL 2026 Poster_

### Official Review · Reviewer_tarq · 2026-01-05

**Confidence:** 4
**Preliminary Rating:** 3
**Final Rating:** 4

**Summary:**

The proposed method aims to be a trainable alternative to slice-by-slice prompting in 2D FM with automatic prompt placement. As 2D FM, ScribblePrompt-SAM, a SAM-based FM, is used. The automatic prompt generation consists of an offset prediction and a refinement module that produces the vessel center of the current slice based on features from the previous slice. It is tested on Vessel Segmentation with a private and a public dataset.

**Strengths:**

* Figure 1 is a nice illustration showing the different paradigms
* Comparison to fully supervised methods, like nnUNet as upper bound.
* detailed description of losses and new modules
* analysis and discussion of performance influence of different components

**Weaknesses:**

* related work for slice-by-slice replacement is not discussed/listed
* alternative methods that reduce the need for the slice-by-slice prompting are not included in the analysis
* motivation with "standard foundation models are inherently 2d" is outdated.

**Detailed Comments:**

in Section 1:
* SAM citation missing
* the references Magg et al 2025, Si et al 2025, Si et al 2024, do not seem to fully support the claims made in the paragraph (maybe the authors can clarify with statements in the papers that lead to the citation)
* since 2025 there are also foundation models that were specifically designed for 3D, eg nnInteractive - this makes the statement "standard foundation models are inherently 2D" outdated

related work: missing other methods that reduce the need for slice-by-slice prompting (training-free and trainable methods) - this would position the proposed method better and ablation studies could show the advantage of this method over other already introduced methods.

**Justification Of Final Rating:**

The authors addressed the major concern of how their work is positioned in the field in the rebuttal and added ablation experiments and comparisons. There are still some minor improvements unaddressed from the rebuttal, but the work presents an interesting idea which could be potentially applied to other SAM-based 2D models.

**Justification Of The Preliminary Rating:**

The method seems sound and is well explained, but some statements need clarification and the work needs to be positioned better compared to related work to show its value and advantages over existing work (see questions to address in the rebuttal).

**Questions To Address In The Rebuttal:**

* Where was the Fe-MRA dataset used in this work collected from?
* Why was ScribblePrompt-SAM chosen over other 2D SAM-based models? And I think it should be made clear earlier on that ScribblePrompt was used. Aside from implementation detail, it is referred to as SAM-based model, then I would like to see also the results for SAM (like for ScribblePrompt - could justify the decision to use ScribblePrompt-SAM over SAM).
* Where are the results of ScribblePrompt-UNet? In the Implementation Details, the SAM-based and UNet-based model are mentioned, but then in the results, there is only one ScribbelPrompt and Ours and it is unclear if this is the SAM-based or UNet-based model.

* Related work, training-free and trainable methods, should be discussed and explain what the advantage of the proposed method is over them.
* How does the proposed method perform compared to training-free methods, such as suggested in μSAM (https://www.nature.com/articles/s41592-024-02580-4) or MedicoSAM (https://ieeexplore.ieee.org/document/11303368) with training-free propagation? (this is related to the position of this work wrt related work).

---

> ### Author Response · Authors · 2026-01-25
> **Response to Reviewer tarq (1/2)**
>
> We thank the reviewer for the thoughtful feedback. We are glad the reviewer appreciated the clarity of our illustrations and the detailed descriptions of our losses and modules. We also value the recognition of our comprehensive component analysis and our inclusion of fully supervised methods like nnU-Net as performance upper bounds. We address the specific questions below:
>
> ### Q1 The details of the in house Fe-MRA dataset
> *Response*:
> The Fe-MRA dataset was retrospectively collected from Nanjing Drum Tower Hospital, The Affiliated Hospital of Nanjing University Medical School, between 2023 and 2025, under an IRB-approved protocol. All patient data were anonymized prior to analysis. We have added these details to the revised manuscript.
>
> It consists of 50 cases exhibiting a wide range of vascular morphologies, including normal vessels, lower extremity varicose veins, and arterial thrombosis. This diversity introduces significant challenges for segmentation, such as severe tortuosity and vessel occlusion.
> The images are whole-body MRA scans acquired 30 minutes post-injection of Ferumoxytol. To ensure data diversity, scans were performed using two different MRI systems: a 1.5T Philips scanner and a 3.0T GE scanner. The dataset features heterogeneous spatial resolutions, with voxel spacing ranging from 0.3x0.3x1.0 mm to 0.8x0.8x1.0 mm, capturing key lower extremity vessels such as the femoral artery/vein and great saphenous vein.
> The ground truth was established through a rigorous two-stage process: initial dense annotation by two junior radiologists, followed by a final review and revision by a senior radiologist to ensure clinical accuracy.
>
> ### Q2 Rationale for selecting ScribblePrompt-SAM as the backbone, clarification on the interaction type (scribble vs. point), and the missing quantitative results for the ScribblePrompt-UNet baseline
> *Response*:
> We thank the reviewer for these insightful questions regarding our backbone selection and experimental baselines. We have revised the manuscript to clarify the usage of ScribblePrompt earlier in the text. Below, we address the specific concerns regarding prompt types, model selection, and the missing UNet results.
>
> 1. Clarification on Prompt Type:
> We explicitly clarify that we strictly used Point Prompts, not Scribble Prompts, in our proposed framework. Although the pre-trained weights we utilized are from the "ScribblePrompt" project, our interaction strategy relies solely on positive clicks to guide the segmentation. This choice was made to simulate the most efficient and common user workflow in clinical settings.
> Even though we utilize point interactions, the ScribblePrompt model was selected as our backbone mainly because: it is extensively trained on a diverse collection of medical imaging datasets. This provides a significantly more robust feature representation for medical structures (like vessels in MRA) compared to the vanilla SAM, which was trained on natural images.
>
> 2. Justification for Choosing ScribblePrompt-SAM over Other Variants:
> The reviewer asks why we chose ScribblePrompt-SAM over other 2D SAM-based models (e.g., MedSAM). The key reasons are twofold: (1) As mentioned, ScribblePrompt's weights are optimized for diverse medical segmentation tasks. (2) Many medical SAM variants (e.g., MedSAM) are heavily optimized for Bounding Box prompts and perform sub-optimally with Point prompts.
> To empirically justify this choice, we conducted additional experiments comparing our backbone with MedSAM. The results, presented in Table 1 below, demonstrate that our ScribblePrompt-based backbone significantly outperforms MedSAM in this point-interaction setting.
>
> Table 1 Comparison with MedSAM on the SEG.A. dataset.
> | Method | Dice (\%) | HD95 (mm) |
> |--------|:---------:|:---------:|
> | MedSAM |   84.57   |   22.47   |
> | Ours   |   86.44   |   19.46   |
>
> 3. Results of ScribblePrompt-UNet:
> Regarding the comparison with UNet, we clarify that the "Ours" results in the main paper are based on the SAM backbone to achieve state-of-the-art performance. To demonstrate the model-agnostic nature of our framework, we conducted additional experiments applying our method to a standard UNet backbone (also using point prompts).
> As shown in Table 2, our method improves performance on the UNet backbone as well, proving that our contribution is effective across different architectures. These ScribblePrompt-UNet and Ours(ScribblePrompt-UNet) results are now explicitly listed in Table 1 of the revised manuscript.
>
>
> Table 2 Comparison of different backbones in ScribblePrompt using Point Prompts.
> | Method         | Backbone | Dice (\%)     | HD95 (mm) |
> |----------------|----------|---------------|-----------|
> | ScribblePrompt | UNet     | 81.55         | 42.39     |
> | Ours           | UNet     | 82.72 (+1.17) | 35.06     |
> | ScribblePrompt | SAM      | 83.00         | 26.42     |
> | Ours           | SAM      | 86.44 (+3.44) | 19.46     |

---

> > ### Comment · Reviewer_tarq · 2026-01-26
> >
> > Thanks to the authors for the provided clarifications and the manuscript changes.
> >
> > regarding Q2, Point 2: I agree with the authors, Med-SAM is only fine-tuned on bounding box input prompts, which makes it an unfit alternative for 2D SAM-based models. However, with the growing number of foundation models, there would have been 2D SAM-based alternatives, also trained on diverse medical segmentation tasks (therefore fitting the criteria named by the authors). Some studies even suggest that medical fine-tuned FMs are not better than SAM, although not trained on a diverse medical dataset. That is also the reason for my comment about testing the method also on SAM instead ScribblePrompt-SAM. The question remains why ScribblePrompt-SAM was chosen as backbone. But I understand, testing for the best backbone is outside of the scope of this work. With the growing number of foundation models, the choice of backbone model just becomes more and more challenging to answer (also for our community in general). The decision can be based on empirical testing or preliminary testing or a "well educated guess". For the authors, it would have been interesting to see that the backbone does not matter (which ScribblePrompt-UNet results already suggest but is not enough evidence).
> >
> > regarding Q3:
> > In Section 1: “Some approaches also attempt to leverage image registration to align adjacent slices for label propagation.” - would be interesting to name some of these approaches with citations.
> > “However, these approaches often lack inherent 3D context and rely heavily on the assumption of minimal inter-slice
> > variation, leading to error accumulation in datasets with anisotropic resolutions or large spacing.” - should probably be after MedicoSAM introduction and seems misplaced.

---

> > > ### Author Response · Authors · 2026-01-28
> > >
> > > Thank you for the constructive discussion and continued engagement. We appreciate the reviewer’s suggestion regarding backbone selection and the comparison between medical fine-tuned models and vanilla SAM. We address the remaining points as follows:
> > >
> > > ### Q2 (Backbone Choice & Vanilla SAM Comparison):
> > > *Response*:
> > > We agree that backbone selection is becoming increasingly challenging and that medical domain fine-tuning is not always a “silver bullet.” To empirically assess the backbone choice, we conducted an additional experiment on the Seg.A. dataset using vanilla SAM (ViT-B).
> > > Consistent with the reviewer’s comment, SAM works quite well. As shown in Table 1, ScribblePrompt-SAM outperforms SAM by an average of 3.82\% (DSC), indicating that the medical adaptation in ScribblePrompt provides a meaningful advantage in our target domain. Importantly, similar trends are also reported in nnInteractive [1] (see Fig. 6 of [1]), where ScribblePrompt achieves the best performance among point-prompted 2D models, providing independent support for this observation.
> > >
> > > Table 1 Comparison with SAM on the SEG.A. dataset.
> > > | Method | Dice (\%) | HD95 (mm) |
> > > |--------|:---------:|:---------:|
> > > | SAM    |   82.62   |   21.31   |
> > > | ScribblePrompt-SAM  |   86.44   |   19.46   |
> > >
> > > We will add this ablation study to the final manuscript. Thank you for your helpful feedback.
> > >
> > > [1] Fabian Isensee, Maximilian Rokuss, Lars Krämer, et al. nnInteractive: Redefining 3d promptable segmentation. arXiv preprint arXiv:2503.08373 (2025).
> > >
> > >
> > > ### Q3 (Citations & Text Structure):
> > > *Response*:
> > > Thank you for the constructive suggestions. We will carefully follow them.
> > >
> > > 1. Citations Added: We have added the recommended citations for registration-based propagation approaches to the related works, including:
> > >
> > >     [1] Adeleh Bitarafan, Mohammad Farid Azampour, Kian Bakhtari, et al. Vol2Flow: Segment 3D volumes using a sequence of registration flows. MICCAI 2022: 609-618.
> > >     [2] Simon Hermann and René Werner. High accuracy optical flow for 3D medical image registration using the census cost function. In Pacific-rim symposium on image and video technology, 2013: 23-35.
> > >     [3] Maria Vakalopoulou, Guillaume Chassagnon, Norbert Bus, et al. Atlasnet: Multi-atlas non-linear deep networks for medical image segmentation. MCCCAI, 2018.
> > >     [4] Guha Balakrishnan, Amy Zhao, Mert R. Sabuncu, et al. Voxelmorph: a learning framework for deformable medical image registration. IEEE TMI, 2019, 38(8): 1788-1800.
> > >     [5] Richard Beare, Bradley Lowekamp, and Ziv Yaniv. Image segmentation, registration and characterization in R with SimpleITK. Journal of statistical software,2018, 86: 1-35.
> > >
> > > 2. Structural Reordering: We fully agree that the sentence regarding "error accumulation" and "lack of inherent 3D context" was misplaced. This limitation is a core characteristic of slice-by-slice propagation frameworks (such as MedicoSAM) rather than registration methods specifically. As suggested, we have relocated this sentence to follow the introduction of MedicoSAM. This reordering improves the logical flow and better situates the motivation for our proposed method.
> > >
> > >
> > > If you have any further questions, please feel free to let us know. We would be glad to provide additional information to support a more informed evaluation of our paper.

---

> > > ### Author Response · Authors · 2026-01-31
> > >
> > > Thank you again for the constructive discussion and the follow-up notes. We wanted to check whether our posted responses and updates address the remaining concerns you highlighted. If anything is still unclear, or if you have any additional questions, we would be happy to clarify further before the discussion period concludes.

---

> ### Author Response · Authors · 2026-01-25
> **Response to Reviewer tarq (2/2)**
>
> ### Q3 Related work regarding training-free and trainable methods and explaining the advantages of the proposed method over them.
> *Response*:
> We appreciate this insightful suggestion. In the revised manuscript, we have significantly expanded the related works to discuss and compare our approach with both training-free and trainable methods.
> 1. Comparison with Training-Free Methods (e.g., $\mu$SAM, MedicoSAM):
> We acknowledge that recent foundation models like $\mu$SAM [1] and Medico-SAM [2] show impressive generalization. However, as noted in the $\mu$SAM paper itself, due to computational constraints, these methods typically process volumetric data "slice by slice" by projecting masks to adjacent slices to derive prompts. This "projection-based" strategy relies on the assumption of minimal variation between slices. It lacks inherent 3D context, leading to error accumulation, especially in datasets with large inter-slice spacing or anisotropic resolutions (as discussed in PAM [4]). In contrast, our method explicitly models 3D inter-slice consistency via local continuity and global consistency losses. As demonstrated in our new ablation study (see Response to Q4), our method outperforms the training-free propagation strategy of $\mu$SAM by 3.16 points in SEG.A., proving that explicit 3D modeling is superior to simple mask projection.
> 2. Comparison with Trainable Methods (e.g., MedSAM2, PAM):
> We further compare our method against state-of-the-art trainable approaches, specifically MedSAM2 [3] and PAM [4]. MedSAM-2 adapts the SAM-2 architecture to medical imaging by employing a memory bank that stores features from neighbor frames. Although this introduces 3D context, the memory mechanism is computationally heavy and primarily designed for video-like continuity. It treats the volume as a temporal sequence, which may not optimally capture the anatomical spatial consistency inherent in static 3D medical volumes. PAM introduces a propagation mechanism where the model utilizes Transformer cross-attention. Specifically, it treats adjacent slice features as Keys ($K$) and Values ($V$) to guide the current slice (Query $Q$). While effective for local propagation, this slice-to-slice attention often struggles with long-range dependencies or when the target structure undergoes significant topological changes that are not captured by immediate neighbors. Unlike MedSAM-2's sequence-based memory or PAM's local attention, our method proposes topology-aware automatic prompt generation together with the local continuity and global consistency losses to explicitly model the structural continuity and consistency without the heavy computational overhead.
>
> We have revised the manuscript to include these citations and the comparative analysis.
>
> [1] Anwai Archit, Luca Freckmann, Sushmita Nair, et al. Segment anything for microscopy. Nature Methods, 2025, 22(3): 579-591.
>
> [2] Anwai Archit, Luca Freckmann, and Constantin Pape. MedicoSAM: Robust Improvement of SAM for Medical Imaging. IEEE Transactions on Medical Imaging, 2025.
>
> [3] Jun Ma, Zongxin Yang, Sumin Kim, et al. Medsam2: Segment anything in 3d medical images and videos. arXiv preprint arXiv:2504.03600, 2025.
>
> [4] Chen, Zifan, Xinyu Nan, Jiazheng Li, et al. PAM: a propagation-based model for segmenting any 3D objects across multi-modal medical images. npj Digital Medicine, 2025.
>
>
> ### Q4 Performance comparison with training-free methods (e.g., $\mu$SAM, MedicoSAM) to clarify the position of this work.
> *Response*:
> Thank you for this valuable suggestion. To address this, we implemented the training-free propagation strategy exactly as described in $\mu$SAM and Medico-SAM, where "segmentation masks are projected to adjacent slices, prompts are derived from them, and segmentation is run for these prompts."
> We compared this baseline against our proposed method. The quantitative results show that the training-free method achieves a SEG.A. score of 83.33, which is 3.11 points lower than our method. While the training-free approach is effective to a certain extent, we observed that it lacks robustness in maintaining continuity across slices, particularly when handling complex topological changes.
>
> In the revised manuscript, we have added a new ablation study experiment. This section compares our method against: (1) The training-free propagation (suggested by you). (2) The registration-based propagation method (suggested by Reviewer XYPn).
> These comparisons demonstrate the effectiveness of our topology-aware automatic prompt generation strategy, which significantly outperforms these baselines by explicitly modeling slice-to-slice consistency.
>
> We hope the rebuttal and revision resolve the main concerns and would be gald to provide additional clarification.

---

### Official Review · Reviewer_ENzh · 2026-01-07

**Confidence:** 5
**Preliminary Rating:** 1
**Final Rating:** 2

**Summary:**

This work investigates whether 3D vessel segmentation can be effectively addressed using a 2D pretrained network, aiming to reduce the reliance on fully 3D models. The authors propose a conceptual strategy to enforce 3D consistency across slices, combining automatic prompt generation (seed-point initialization for the pretrained model) with two additional loss functions designed to preserve spatial continuity and contextual coherence. The approach is evaluated on two datasets, one internal and one external.

**Strengths:**

The paper explores a lightweight and potentially more data-efficient alternative to 3D segmentation networks, which is highly relevant for medical imaging applications with limited annotations or computational resources. The attempt to enforce 3D consistency with specific losses and using a 2D pretrained model is conceptually interesting. Although the comparison with supervised 3D methods is mainly numerical, it still provides a useful reference point.

**Weaknesses:**

A key weakness of the paper is that the conceptual framework is not sufficiently supported by a clear and trustworthy implementation, making it difficult to assess whether the proposed ideas can be reliably reproduced. In particular, it is unclear how training is performed, if at all: the paper does not adequately specify which components are trained, which are frozen, and how the pretrained 2D network is integrated into the overall pipeline. This lack of clarity raises concerns about the methodological soundness of the approach.
While the two proposed loss functions are conceptually interesting, their practical implementation is not described in enough detail to allow replication or to understand their actual contribution to performance. Similarly, the enforcement of 3D consistency remains largely at a high-level description, without sufficient algorithmic or architectural detail.

**Detailed Comments:**

•	The paper describes the architecture as a 3D Network, but you use a “pre-trained 2D SAM encoder”, meaning it’s 2D, reinforced by the statement: “Given a 3D volume, we process it as a sequence of 2D slices”. Stating is a 3D Network, like in Figure 1, it looks incorrect to me.
•	The Local Continuity Loss is conceptualised to penalise changes in the morphology, but it’s not clear where you apply it, at which stage? Since the 2D network is already trained.
•	The Global Consistency Loss is unclear about what it applies to. The paper reports feature embeddings F on the feature map extracted by the Prompt Encoder, but it doesn’t provide sufficient detail. Also, more motivation is needed to justify conceptually why this should work; semantic consistency is not enough.
•	Not specified how the training and testing sets are split. By volumes or by slices? Is there any training involved? As specified in the Implementation details section.
•	Figure 3: I would specify, with arrows in the third-row plot, the slices displayed in rows 1 and 2.
•	Figure 3: Please also include supervised method outcomes.
•	Figure 3 clearly shows the method’s limits: it can segment only one connected component per slice. Something a 3D network would easily address.

**Justification Of Final Rating:**

The additional training and architectural details provided in the rebuttal improve the clarity of the framework and make the overall approach easier to assess. However, some issues remain unresolved. In particular, the presence of a “3D Network” in Figure 1 is still misleading and inconsistent with the proposed 2D-based formulation. The description and empirical validation of the two proposed loss components are still insufficient, limiting confidence in their effectiveness. Finally, despite the authors’ statement that visual comparisons with nnU-Net would be included in Figure 3, these results are not provided, which weakens the comparative evaluation.

**Justification Of The Preliminary Rating:**

Despite an interesting conceptual idea, the method lacks sufficient technical clarity and implementation detail, making it difficult to assess reproducibility and practical feasibility. In addition, addressing a 3D segmentation problem with a 2D network is inherently problematic unless 3D consistency is convincingly resolved, which is not demonstrated here. Overall, the approach appears premature, and its potential impact is not adequately supported by the current evidence.

**Questions To Address In The Rebuttal:**

I would focus on explaining the architecture in the rebuttal, without assuming a deep knowledge of SAMed. See above.

---

> ### Author Response · Authors · 2026-01-25
> **Response to Reviewer ENzh (1/2)**
>
> We thank the reviewer for the constructive feedback. We are encouraged that the reviewer highlighted our method as a "lightweight and data-efficient alternative" to full 3D networks, which is highly relevant for clinical settings with limited resources. We also appreciate the positive remarks regarding our "conceptually interesting" approach to enforcing 3D consistency using a 2D pretrained foundation. We take the concerns about technical clarity seriously. We will release all implementations upon the acceptance of the work.
>
> Beyond efficiency, we would like to emphasize that our framework’s practical feasibility, specifically the "single point for whole volume" interaction paradigm, and our topology-aware evaluation. These components work together to ensure that our method is not only computationally efficient but also anatomically accurate and easy to deploy.
>
> In the revision, we focused on making the method description crystal clear: we clarified (i) what is trained vs. frozen, (ii) how the 3D consistency losses are computed and where gradients flow, and (iii) how train/val/test are split at the patient (volume) level (Sec. 3.2 and Sec. 3.1). We hope the point-by-point clarifications below address your concerns.
>
> ### Q1 Architecture \& Training Details
> *Response*: We appreciate the reviewer pointing out the need for greater clarity regarding the network architecture and training details. We confirm that training is indeed a core part of our method, and our framework is built upon a robust, medically pre-trained foundation.
> To ensure reproducibility, we provide a detailed breakdown of the architecture and training protocol below and revised Sec. 3.2 to explicitly specify the training configuration (without assuming prior knowledge of SAMed):
>
>
> 1 Architecture Overview (Frozen vs. Trainable):
> Our pipeline processes a 3D volume as a sequence of 2D slices ($I_1, I_2, ..., I_N$). The network consists of three main modules with distinct training statuses:
> - Image Encoder (Frozen) and Prompt Encoder (Frozen): We utilize the ViT-b backbone from ScribblePrompt-SAM as the Image Encoder. Since ScribblePrompt-SAM has already been extensively pre-trained on a large-scale collection of medical images, its feature extraction capabilities are well-aligned with our task. We keep this component completely frozen during training to preserve these robust medical features and maintain computational efficiency. It extracts image embeddings for each 2D slice. The standard Prompt Encoder is also frozen, as it provides positional encodings for point prompts that do not require domain-specific adaptation.
> - Automatic Prompt Generator (Trainable): This is our core contribution, a fully trainable module designed to bridge the gap between 2D slice processing and 3D volumetric consistency. It consists of a lightweight Offset Prediction Network (MLP) that takes the feature embedding from the previous slice ($F_{i-1}$) to predict the vessel's trajectory offset. This module is trained end-to-end to learn robust slice-to-slice prompt propagation under the vessel topology and appearance changes, transforming the 2D foundation model into a semi-automatic 3D tracking tool without requiring manual prompts for every slice.
> - Mask Decoder (Trainable): We fine-tune the Mask Decoder. Although initialized from ScribblePrompt-SAM, fine-tuning is necessary to adapt the model to the specific task of vessel segmentation, which requires high sensitivity to thin, tubular structures that may differ from the general "scribble-based" segmentation targets seen during pre-training.
>
> 2 How Training is Performed: The training process is iterative and involves the following steps for a given input volume:
> - Forward Pass: The frozen encoder produces embeddings for the current slice. The trainable Prompt Generator creates prompts, and the Mask Decoder predicts the mask.
> - Loss Calculation: Standard segmentation Dice loss is computed against the 2D Ground Truth. The Local Continuity Loss and Global Consistency Loss are calculated by comparing the current slice's predictions/features with those of adjacent slices.
> - Backward Pass: Since the Image Encoder is frozen, gradients from all losses (including the 3D consistency losses) are back-propagated exclusively to update the Prompt Generator and the Mask Decoder.
>
> In the Training phase, we employ a randomized adjacent-slice sampling strategy. In each training iteration, we randomly sample pairs of adjacent slices $(I_{i-1}, I_i)$ from the training volumes. The model takes the prediction/features from $I_{i-1}$ to predict the prompt for $I_i$. This forces the Prompt Generator to learn the robust slice-to-slice transition logic rather than memorizing specific volumetric positions.

---

> ### Author Response · Authors · 2026-01-25
> **Response to Reviewer ENzh (2/2)**
>
> 3 Data Splitting:
> We strictly perform data splitting by Volume, not by slices.
> - For Fe-MRA Dataset: 50 patients were split into Training (30), Validation (5), and Testing (15) at the patient level.
> - For SEG.A Dataset: We followed the official challenge split guidelines (patient-level). This ensures there is no data leakage between training and testing slices, guaranteeing a fair evaluation of the model's generalization capability.
>
>
> ### Q2 Calling it a “3D network” if it processes 2D slices
> *Response*:
> We appreciate the comment and agree that our original wording could be misleading. While we use a 2D network as the backbone, we introduce a learnable module that explicitly models interactions between adjacent slices, and the network processes two neighboring slices jointly at each step (input shape: 2xHxW), which also does not fit the definition of a 2D network. To avoid confusion, we refrain from describing our method as a “2D or 3D network,” and instead define it as a slice-wise framework built on a 2D promptable backbone with explicit 3D consistency constraints and inter-slice learning.
>
> ### Q3 For the loss functions ($L_{local}$ and $L_{global}$), where are these applied and why do they work
> *Response*:
> Both losses are applied between adjacent slices, which is central to our design.
>
> With a two-slice input, the model outputs two probability maps, $P_i$ and $P_{i-1}$. The local continuity loss $L_{local}$ is applied between them, directly optimizing the Mask Decoder to enforce continuous vessel boundaries across neighboring slices and penalize abrupt shape changes.
>
> The global consistency loss $L_{global}$ is applied to the corresponding two feature embeddings. It encourages the Prompt Generator to learn semantic consistency of vessel-center representations across slices, guiding it to track the same anatomical structure.
>
> Together, these losses model inter-slice relationships, motivating our two-slice input and cross-slice learning. As a result, the model better captures 3D anatomical continuity while maintaining the efficiency of a 2D backbone. We clarified these loss definitions and where they are applied in the revised Implementation Details (Sec. 3.2).
>
>
> ### Q4 Topological Limitations (Single Component)
> *Response*:
> We agree that single-point prompting can be challenged by multi-branch or multi-instance cases (e.g., separating multiple disconnected components within a slice). We believe both 2D and 3D models have their strengths and applications, particularly when considering computational cost and annotation burden. Recent works such as MedSAM2 demonstrate the broad impact and continued development of slice-wise models, indicating that 2D-based approaches remain an active and relevant research direction rather than a superseded one.
>
> Our current design targets both practical and common clinical scenarios involving continuous anatomical structures, such as tracing a main vessel tree (e.g., the aorta or major peripheral arteries). By leveraging cross-slice learning with a 2D backbone, the method offers a favorable trade-off between 2D and 3D models in terms of accuracy, computation, and annotation cost. In the revision, we explicitly acknowledge these scope/limitation points and outline extensions such as multi-hypothesis tracking and/or multiple prompts for complex branching.
>
> ### Q5 Figures
> *Response*:
> We thank the reviewer for the suggestion. In the revision and the final version, we will:
> 1. Add explicit arrows in Figure 3 to indicate the slice correspondence.
> 2. Include visual outputs from nnU-Net (Supervised) in Figure 3 for a direct visual comparison.
> 3. Update the caption to clearly describe the slice-wise propagation visualization.
>
> We thank the reviewer again for pushing us to improve clarity. We hope the rebuttal and revision resolve the main concerns and would be glad to provide additional clarification.

---

### Official Review · Reviewer_XYPn · 2026-01-12

**Confidence:** 4
**Preliminary Rating:** 4
**Final Rating:** 5

**Summary:**

This paper proposes a practical framework to “unlock” a pre-trained 2D promptable foundation model (SAM-family) for 3D vessel segmentation by automatic prompt generation. Starting from a single user-provided point prompt on an initial slice, the method propagates prompts slice-by-slice using (i) a feature-driven offset prediction and (ii) a confidence-guided refinement. To reduce slice-wise inconsistencies and topological breaks, the authors also introduce local continuity regularization on probability maps and a global consistency loss in the latent feature space. Experiments on a public CTA dataset (SEG.A) and an in-house Fe-MRA dataset show improved Dice/clDice and reduced topological errors compared to other point-prompt baselines.

**Strengths:**

* Practical motivation and feasibility: Leveraging an existing 2D SAM-style model to handle 3D volumes is appealing in settings where training a full 3D foundation model is expensive or impractical. The “single point for whole volume” interaction paradigm is compelling (Fig. 1)

* Clear technical decomposition: The framework is easy to follow: geometric constraints (local/global) for volumetric coherence + automatic prompt propagation via offset/refinement (Sec. 2, Fig. 2).

* Topology-aware evaluation: Including connectivity-oriented metrics (clDice, Betti error) strengthens the empirical validation for vessels, beyond Dice alone (Sec. 3).

* Ablations are informative: The ablation table suggests each component contributes incrementally, with refinement reducing error propagation and improving topology metrics (Table 3).

**Weaknesses:**

* Single-axis slice processing may limit robustness and 3D coherence.

* Offset prediction is not obviously vessel-specific and lacks comparison to registration-based tracking.

* Task scope is narrow; generality beyond vessels is not demonstrated.

**Detailed Comments:**

### (1) Single-axis slice processing may limit robustness and 3D coherence.
The current pipeline processes the volume as a sequence of 2D slices and enforces continuity along that slice dimension (e.g., the gradient consistency along the slice axis and adjacent-slice feature similarity). While effective, it is unclear whether restricting propagation to a single axis is optimal for volumes with anisotropic spacing, tortuous anatomy, or imaging artifacts.

### (2) Offset prediction is not obviously vessel-specific and lacks comparison to registration-based tracking.
The offset module uses feature space information from the previous slice to predict a displacement (Eq. 4) and then refines using a confidence peak (Eq. 5). While described as learning vessel trajectory/branching, the mechanism appears broadly applicable to any slowly varying structure and is not explicitly constrained by tubular geometry. A natural baseline is inter-slice image registration (rigid/affine or lightweight deformable) to transfer prompt location between adjacent slices; it may provide improved localization at the cost of speed. The paper would benefit from either (i) a comparison to registration-based prompt propagation or (ii) justification why the proposed offset approach is preferable beyond runtime.

### (3) Task scope is narrow; generality beyond vessels is not demonstrated.
Although the method is framed as generally enabling 2D promptable models for 3D segmentation, experiments focus on vessel segmentation (CTA aortic tree; Fe-MRA peripheral vessels). It remains unclear how well the automatic prompting and continuity losses transfer to other common 3D tasks (organs, tumors, lesions) where topology/shape evolution may differ (e.g., non-tubular structures, large appearance changes across slices).

**Justification Of Final Rating:**

The authors have provided sufficient evidence in their extended experiments to show the effectiveness of the proposed method and fully addressed my concerns. I think the current manuscript is solid enough for more insight discussion in the conference.

**Justification Of The Preliminary Rating:**

I lean toward Weak Accept because the paper presents a practical and timely solution to a common bottleneck in 3D medical image segmentation: generating usable 3D masks without the heavy cost of training and deploying 3D foundation models. The core contribution—enabling 3D coarse label generation by propagating prompts and enforcing volumetric consistency while leveraging an off-the-shelf 2D SAM-style model—is both feasible and potentially impactful for real clinical/research pipelines where annotation budgets are limited and compute constraints are real.

**Questions To Address In The Rebuttal:**

* Multi-axis propagation: Have the authors experimented with applying the same prompt propagation strategy along all three axes (axial/sagittal/coronal) and fusing results (e.g., majority vote, uncertainty-weighted fusion, or stitching in 3D)? If not, can they discuss expected benefits/limitations and whether single-axis processing was chosen for efficiency or due to data anisotropy?

* Registration baseline: Can the authors compare the proposed feature-driven offset prediction to a simple inter-slice registration baseline for prompt localization (e.g., register slice i−1 to i and warp the prompt/center)? Even a lightweight method would clarify whether the learned offset provides superior robustness (especially near bifurcations, low-contrast regions) rather than simply being a faster heuristic.

* Generality to other 3D segmentation tasks: Do the authors expect the method to work for non-vascular 3D segmentation (organs, tumors, airways, etc.)? If so, can they provide either (i) preliminary results on at least one additional task/dataset or (ii) a discussion of what components would need modification (e.g., prompt type, refinement strategy, losses) when the target is not tubular or changes abruptly across slices?

---

> ### Author Response · Authors · 2026-01-25
> **Response to Reviewer XYPn (1/2)**
>
> We thank the reviewer for recognizing the practical value of leveraging a promptable 2D foundation model for 3D volumes and for appreciating the single-point-for-whole-volume interaction paradigm, the clear modular design, and the topology-aware evaluation. We respond to the specific questions below.
>
> ### Q1: Multi-axis propagation vs. Single-axis processing
> *Response*:
> We appreciate the reviewer's suggestion regarding multi-axis fusion. While multi-view fusion is beneficial in general 3D segmentation, we chose the single-axis (Z-axis) strategy based on two key observations: anatomical structure and inter-slice continuity.
>
> 1. Anatomical Suitability for Vessels: our primary targets (aorta and lower limb arteries) are tubular structures running longitudinally along the body: (1) Slicing along the Z-axis yields vascular cross-sections (circular/elliptical shapes). SAM is highly robust at segmenting these compact shapes given a center point. (2) Slicing along other axes results in long, thin, and often disconnected vascular strips due to tortuosity. Tracking a center point becomes ill-defined and prone to failure when the vessel curves out of the imaging plane. Therefore, the Z-axis is the most geometrically reliable direction for tracking these vessels.
>
> 2. Empirical Validation on General Organs (Abd-CT dataset): to verify if this observation holds for non-tubular organs (addressing the generality concern in Q3), we conducted an ablation study on the Abd-CT dataset [1] (including spleen, kidneys and liver) comparing propagation along different axes.
>
> As shown in Table 1, the Z-axis (Axial) consistently yields the best performance. This is consistent with the common anisotropic resolution of clinical scans (higher in-plane resolution than through-plane), where axial adjacency provides more reliable slice-to-slice matching for our prompt propagation module.
>
> Table 1 Comparison of propagation performance along different axes on Abd-CT.
> |              | Spleen | Right Kidney | Left Kidney | Liver |  Mean |
> |--------------|:------:|:------------:|:-----------:|:-----:|:-----:|
> | Axial (Z)    |  79.60 |     85.54    |    86.26    | 80.45 | 82.96 |
> | Sagittal (X) |  72.53 |     46.02    |    47.11    | 16.96 | 45.66 |
> | Coronal (Y)  |  64.88 |     66.16    |    39.46    | 38.59 | 52.27 |
>
> [1] Bennett Landman, Zhoubing Xu, Juan Igelsias, Martin Styner, Thomas Langerak, and Arno Klein. Miccai multi-atlas labeling beyond the cranial vault–workshop and challenge. In MICCAI workshop, page 12, 2015.

---

> ### Author Response · Authors · 2026-01-25
> **Response to Reviewer XYPn (2/2)**
>
> ### Q2: Comparison with traditional registration-based propagation (Rigid/Affine)
> *Response*: We appreciate this insightful suggestion. To verify the necessity of our learning-based prompt generation strategy, we conducted additional experiments using traditional registration methods provided by SimpleITK. We tested both Rigid and Affine transformations (initialized by mask centers/centroids) to propagate masks between adjacent slices.
> The quantitative results are presented in Table 2 below. As observed, the best registration baseline (Affine Center) lags behind our method by 2.02\% in Dice score. While registration provides a reasonable baseline, this 2\% gap is critical in vascular segmentation. Traditional registration often fails to capture non-linear deformations and rapid topological changes of vessels (e.g., bifurcations), whereas our deep learning-based method learns to predict these complex boundary evolutions accurately.
>
> Table 2 Comparison with Registration-based Methods (Dice Score, \%).
> | Method            |   Dice Score   | Inference Time (s/slice) |
> |-------------------|:--------------:|:------------------------:|
> | Rigid (centroid)  | 59.29 (-27.15) |           0.55           |
> | Rigid (center)    | 54.21 (-32.23) |           0.55           |
> | Affine (centroid) |  82.57 (-3.87) |           0.56           |
> | Affine (center)   |  84.42 (-2.02) |           0.56           |
> | Our               |      86.44     |           0.10           |
>
> A major limitation of traditional registration is computational cost. SimpleITK-based registration requires iterative optimization for every slice pair, which is computationally expensive. In contrast, our method operates in a feed-forward manner, offering significantly faster inference speeds suitable for real-time clinical interaction.
>
> We acknowledge that the performance of registration methods heavily depends on hyperparameter tuning (e.g., learning rate, iterations). Due to the limited time during the rebuttal phase, the results in Table 2 represent our best effort with standard settings. As suggested, we will further optimize the registration baselines and include a more comprehensive comparison in the final version of the paper.
>
> We acknowledge that the performance of registration methods can depend on hyperparameter tuning (e.g., step size, iterations). We used standard settings consistent with common SimpleITK practice; the results in Table 2 reflect this controlled baseline comparison.
>
> In summary, while registration is a valid approach, our method offers a superior trade-off between higher segmentation accuracy and inference efficiency.
>
>
> ### Q3: Generalizability of the proposed method to non-vascular structures
> *Response*: We thank the reviewer for raising the important question regarding the generalizability of our method. While our initial focus was on vascular structures (aorta and lower limb arteries) due to their complex topology, the core premise of our framework, which leverages inter-slice spatial consistency for mask propagation, is applicable to general volumetric segmentation tasks.
>
> To demonstrate this, we conducted additional experiments on the Abd-CT dataset, targeting three major abdominal organs: Spleen, Kidneys and Liver. These organs represent "blob-like" structures, which pose different challenges compared to tubular vessels.
>
> We applied our method (using the Z-axis propagation strategy validated in Q1) to these organs. The results are summarized in Table 3 below.
>
> Table 3 Segmentation performance (Dice Score, \%) on AbdCT dataset.
> | Method | Spleen | Right Kidney | Left Kidney | Liver |  Mean |
> |--------|:------:|:------------:|:-----------:|:-----:|:-----:|
> | nnUNet |  90.83 |     89.39    |    86.75    | 95.57 | 90.64 |
> | Our    |  79.60 |     85.54    |    86.26    | 80.45 | 82.96 |
>
> As shown in Table 3 , our method achieves robust segmentation performance across all three organs. This success indicates that our automatic prompt generation module is robust to shape variations, effectively handling the slice-by-slice shape transitions of blob-like organs in a manner similar to vascular. Furthermore, these results demonstrate the versatility of our approach, confirming that the proposed automatic prompt generation is not overfitted to circular vascular shapes but instead learns general boundary deformation features.
>
> We will include qualitative visualization results of these abdominal organs in the final version, showing that our method can generate smooth and accurate 3D masks for general organs without topological breaks.
>
> In conclusion, our method is not limited to vascular segmentation but serves as a generalizable solution for volumetric medical image segmentation. We hope the rebuttal and revision resolve the main concerns and would be gald to provide additional clarification.

---

> > ### Comment · Reviewer_XYPn · 2026-01-30
> >
> > Thank you for your response with all these supplemental experiment results that clarify most of my concerns. This work looks very solid to me.

---

> > > ### Author Response · Authors · 2026-01-31
> > >
> > > Thank you for the encouraging feedback. We appreciate your time and careful reading, and we are glad the additional experiments helped clarify your concerns. If any further questions come up during the discussion period, we would be happy to provide clarification.

---

### Author Rebuttal · Authors · 2026-01-25

**Rebuttal:**

We sincerely thank the reviewers for their constructive feedback and for recognizing the novelty and practical value of our work. We are encouraged that **Reviewer XYPn** found the framework “easy to follow” and the “single-point-for-whole-volume” interaction paradigm “compelling,” and appreciated our topology-aware evaluation (clDice / Betti). We also appreciate **Reviewer ENzh**’s emphasis on “technical clarity” and reproducibility, which prompted us to make the training setup and frozen/trainable components explicit in the revision. We thank **Reviewer tarq** for highlighting the importance of positioning relative to recent interactive and propagation-based baselines, which we address in the revision and point-by-point responses.

### Rebuttal highlights
- **Training clarified**: we now explicitly state which components are **frozen vs. trainable**, and provide the adjacent-slice training protocol and gradient flow (**Sec. 3.2**).
- **Evaluation**: we clarify that splitting is performed strictly at the **patient / volume** level (not by slices) for Fe-MRA and SEG.A (**Sec. 3.1**).
- **New propagation baselines**: we add comparisons to **rigid / affine registration-based propagation** and a **training-free $\mu$SAM-style** baseline, and report **per-slice inference time** (**Table 4**).
- **Scope and limitations clarified**: we expand the limitations discussion on slice-wise error accumulation and challenging turn-back / branching cases, and outline concrete future directions (bi-directional propagation; multi-hypothesis tracking) (**Sec. 4**).
- **Multi-axis**: we include a small **axis-wise propagation** study on Abd-CT.
- **Key empirical takeaway**: across both datasets, the rebuttal/revision makes clear that our gains are not only in Dice but also in **connectivity/topology** (higher clDice; lower Betti error), while remaining **competitive with fully supervised upper bounds** with orders-of-magnitude lower interaction.

We hope these revisions and clarifications address concerns on clarity and positioning. We respond to each reviewer point-by-point below.

**Supporting Material:**

/attachment/3e1073078e69f44b015eb149cc039b0b69665b29.pdf

---

### Meta-Review · Area_Chair_i9c2 · 2026-02-03

**Recommendation:** Accept (Poster)
**Confidence:** 4

**Metareview:**

The reviewers recognised the contribution and improved clarity in the rebuttal stage. Though there are still some unclear parts of the model structure, comparisons with the widely used nn-UNet are to be included. Due to the limited pages and short discussion period, it is acceptable if the novelty is attractive. In general, an interesting paper, and worth discussion in conference.

---

### Decision · Program_Chairs · 2026-02-13

Accept (Poster)